# Design of Clofazimine-Loaded Lipid Nanoparticles Using Smart Pharmaceutical Technology Approaches

**DOI:** 10.3390/pharmaceutics17070873

**Published:** 2025-07-02

**Authors:** Helena Rouco, Nicola Filippo Virzì, Carolina Menéndez-Rodríguez, Carmen Potel, Patricia Diaz-Rodriguez, Mariana Landin

**Affiliations:** 1Departamento de Farmacología, Farmacia y Tecnología Farmacéutica, Grupo I+D Farma (GI-1645), Facultad de Farmacia, Universidade de Santiago de Compostela, 15782 Santiago de Compostela, Spain; patricia.diaz.rodriguez@usc.es; 2Instituto de Investigación Sanitaria de Santiago de Compostela (IDIS), 15706 Santiago de Compostela, Spain; 3Instituto de Materiais da Universidade de Santiago de Compostela (iMATUS), 15706 Santiago de Compostela, Spain; 4Department of Drug and Health Science, University of Catania, Viale A. Doria 6, 95125 Catania, Italy; nicola.virzi@phd.unict.it; 5Microbiology Laboratory, Complexo Hospitalario Universitario de Vigo, 36312 Vigo, Spain; carolina.menendez.rodriguez@sergas.es (C.M.-R.); carmen.potel.alvarellos@sergas.es (C.P.)

**Keywords:** nanostructured lipid carriers, clofazimine, drug delivery, artificial intelligence, docking

## Abstract

**Background/Objectives**: Clofazimine (CFZ) is a versatile antimicrobial active against several bacterial species, although its reduced aqueous solubility and the occurrence of side effects limit its use. Nanostructured lipid carriers (NLCs) constitute an interesting approach to increase drug bioavailability and safety. However, the development of nanoparticle-based formulations is challenging. In the present work, a combination of smart pharmaceutical technology approaches was proposed to develop CFZ-loaded NLCs, taking advantage of previous knowledge on NLCs screening. **Methods**: A design space previously established using Artificial Intelligence (AI) tools was applied to develop CFZ-loaded NLC formulations. After formulation characterization, Neurofuzzy Logic (NFL) and in silico docking simulations were employed to enhance the understanding of lipid nanocarriers. Then, the performance of formulations designed following NFL guidelines was characterized in terms of biocompatibility, using murine fibroblasts, and antimicrobial activity against several strains of *Staphylococcus aureus*. **Results**: The followed approach enabled CFZ-loaded NLC formulations with optimal properties, including small size and high antimicrobial payload. NFL was useful to investigate the existing interactions between NLC components and homogenization conditions, that influence CFZ-loaded NLCs’ final properties. Also, in silico docking simulations were successfully applied to examine interactions and affinity between the drug and the lipid matrix components. Finally, the designed CFZ-loaded formulations demonstrated suitable biocompatibility, together with antimicrobial activity. **Conclusions**: The implementation of smart strategies during nanoparticle-based therapeutics development, such as those described in this manuscript, would enable the more efficient design of new systems for suitable antimicrobial delivery.

## 1. Introduction

Clofazimine (CFZ) is classified as a group-B medication by the WHO (World Health Organization) and is recommended for the treatment of drug- and multidrug-resistant tuberculosis [1]. It has also been used as a first-line drug to treat multibacillary disease caused by *Mycobacterium leprae* [2]. Moreover, recent studies showed the triple antibiotic combination of CFZ, rifabutin, and clarithromycin is effective against biofilms and airway infections caused by *Mycobacterium avium* subspecies *hominissuis* [3]. Furthermore, in addition to its widely known antimycobacterial effect, CFZ is also active against *Staphylococcus aureus*, including methicillin-resistant strains (MRSA), which have become a major health concern in recent years [4]. However, despite its versatility, the low aqueous solubility and high lipophilicity of clofazimine limit its absorption after oral administration, requiring the use of higher antibiotic doses [1]. Unfortunately, oral administration of elevated clofazimine doses often leads to side effects, such as gastrointestinal damage or skin discoloration [1]. Interestingly, recent findings claim that nanotechnology offers great potential to re-explore the capabilities of already known antimicrobial compounds, such as CFZ [5]. In this way, nanoparticle-based drug delivery systems can efficiently increase the aqueous solubility of hydrophobic antibiotics and antimicrobial compounds’ half-life. Moreover, they can provide passive targeting to phagocytic cells, especially useful to treat intracellular infections [5,6]. All these features may allow reducing the required dose to achieve the desired therapeutic effect, in comparison with free antimicrobial compounds, decreasing the risk of side effects and toxicity [7]. Amid the existing nanoparticulated systems, Nanostructured lipid carriers (NLCs), known as the second generation of Lipid Nanoparticles (LNs) [8], constitute an excellent option to formulate hydrophobic antimicrobial compounds. Their hybrid matrix integrating both solid and liquid lipids offers several advantages over the previous generation of LNs. Among them, increased drug-loading capacity, higher stability, and/or ability to modulate drug release [8,9,10]. Indeed, these systems have proven utility for the encapsulation of highly hydrophobic antimicrobials such as rifabutin (RFB), achieving controlled release and high loading capacity [11,12]. Furthermore, these colloidal systems can be composed of biodegradable and GRAS (Generally Recognized As Safe) materials, making them suitable to develop a safe alternative to conventional antibiotic formulations [10].

Despite the promising NLC characteristics, the development of optimized nanoparticle-based drug delivery systems can be a challenging task, due to the high number of variables involved. Nanoparticles are complex structures formed by several components where all of them control the final physicochemical and stability properties of the systems. Therefore, their characteristics are modified by subtle modifications in composition and operation conditions [13]. Considering the above, a comprehensive understanding of the critical components and their interactions is mandatory to obtain reproducible nanocarriers with optimal characteristics [13].

In recent years, Artificial Intelligence (AI) has emerged as a powerful tool to guide and optimize the development of pharmaceutical products [14]. Among the different AI tools available, the application of Neurofuzzy Logic (NFL) in pharmaceutical design enables us to understand the effect of composition and operation conditions on the characteristics of the obtained formulations [11]. NFL is a hybrid technology that integrates the capabilities of Artificial Neural Networks (ANNs) and Fuzzy Logic [15]. On the one hand, ANNs enable us to detect data trends and relationships, and to learn from experience, mimicking the learning process of the human brain [16]. On the other hand, Fuzzy Logic allows us to express concepts in a simple way, through the generation of “IF-THEN” rules [15].

The effectiveness of NFL in guiding the development of a wide variety of pharmaceutical products has already been proven. As an example, this technology has successfully been applied to optimize process parameters during the development of core–shell microparticles [17], thermosensitive hydrogels [18], and polymeric nanoparticles [19]. Furthermore, in a previous work in our laboratory, NFL was successfully employed to assist in the development of an RFB-loaded NLCs formulation [11]. The followed approach enabled us to delimit the design space of the nanocarriers [11], defined as the multidimensional combination of process parameters and input variables that ensure product quality [20].

Building on this foundation, the aim of this study is to apply the previously established RFB-loaded NLCs’ design space to the development of CFZ-loaded formulations. This approach shows several benefits, minimizing the formulation development time and cost, together with higher regulatory flexibility [21]. Furthermore, NFL will also be applied after the NLCs characterization to assess existing interactions between formulation parameters and their impact over nanoparticle characteristics after CFZ encapsulation. Additionally, in silico docking will be employed to understand the interactions between clofazimine and lipid matrix components at the molecular level. The use of these smart technologies will provide a better understanding of the complexity of antibiotic-loaded NLC formulations, allowing us to highly efficiently develop suitable CFZ-loaded NLCs. Finally, the biological activity of the obtained formulations will be characterized in terms of biocompatibility, and antimicrobial activity against several strains of *Staphylococcus aureus*, including methicillin and multidrug-resistant ones.

## 2. Materials and Methods

### 2.1. Materials

Precirol^®^ ATO 5 (solid lipid, SL) was kindly donated by Gattefossé (Saint-Priest, France). Oleic acid laboratory reagent grade (liquid lipid, LL) was purchased from Fisher Scientific (Waltham, MA, USA). Polysorbate 80 (Tween^®^ 80) and dimethyl sulfoxide (DMSO) were acquired from Sigma Aldrich (Taufkirchen, Germany). Lecithin (Epikuron^®^ 145 V) was obtained from Cargill (Barcelona, Spain). Clofazimine was purchased from Santa Cruz Biotechnology (Dallas, TX, USA). Dimethyl formamide (DMF) was purchased from Merck Millipore (Burlington, MA, USA). Sterile filter paper discs, Dulbecco’s Modified Eagle Medium (DMEM), foetal bovine serum (FBS), antibiotic-antimycotic, and Alamar Blue™ were acquired from Thermo Fisher Scientific (Waltham, MA, USA). Petri dishes with Mueller-Hinton (MH) agar and sterile saline solution were obtained from BioMérieux (Marcy-l’Étoile, France). BALB/3T3 murine fibroblasts cell line was purchased from ATCC (Manassas, VA, USA). MilliQ^®^ water (Millipore Ibérica, Navalafuente, Spain) was used throughout.

### 2.2. Clofazimine-Loaded NLCs Formulation

NLC formulations were prepared by hot high shear homogenization following a previously described protocol, and a previously established formulation design space [11]. In brief, 300 mg of a lipid phase consisting of a blend of LL, SL, and CFZ, as specified in Table 1, was prepared. Then, an aqueous phase made of 10 mL of MilliQ^®^ water, containing Tween 80^®^ and lecithin in the proportions detailed in Table 1, was also prepared. Afterwards, both phases were heated at 80 °C in a water bath and the aqueous phase was added onto the lipid phase. The mixture was then homogenized, using the speeds indicated in Table 1, for 10 min using an Ultra-Turrax T25 (IKA Labortechnik, Staufen, Germany) while keeping the temperature constant. Finally, CFZ-loaded NLC dispersions were allowed to cool in an ice bath. Each formulation was carried out in duplicate. Blank formulations without CFZ were also prepared as a control, following the above-mentioned procedure.

### 2.3. Clofazimine-Loaded NLCs Characterization

#### 2.3.1. Particle Size, Surface Charge, and Polydispersity

Clofazimine-loaded NLC formulations’ particle size, polydispersity index (PdI), and zeta potential (ZP) were measured in a Zetasizer Nano-ZS (Malvern Instruments, Malvern, UK). Particle size and polydispersity index determinations were performed through dynamic light scattering (DLS) using polystyrene cuvettes (DTS 50012, Malvern Instruments, Malvern, UK) after proper dilution with MilliQ^®^ water (1:100). On the other hand, nanoparticles’ surface charge was determined through particle mobility in an electric field to calculate zeta potential (ZP). Samples were also diluted in MilliQ^®^ water (1:100) and measurements were carried out in a DTS 1070 cuvette (Malvern Instruments, Malvern, UK), where a potential of ±150 mV was established. All the measurements were performed in triplicate for each formulation.

#### 2.3.2. Encapsulation Efficiency and Drug-Loading Quantification

To evaluate the capacity of the developed NLCs to incorporate clofazimine, the encapsulation efficiency (EE%) and the drug loading (DL%) of the different formulations were determined. Towards this goal, freshly prepared CFZ-loaded NLC formulations were first dialyzed using a Spectra/Por^®^ 3 dialysis membrane (MWCO: 3.5 kDa, Fisher Scientific, Waltham, MA, USA), to remove the free drug from the nanoparticle dispersion. Afterwards, formulations were mixed with DMF in a 1:5 ratio to dissolve the NLCs’ lipid matrix, extracting the encapsulated drug. Then, the tubes containing the NLCs: DMF mixture were centrifuged at 15,295 × *g* and 4 °C for 30 min, to precipitate the lipids, while the extracted drug remained on the supernatant. After that, the drug concentration in the supernatant was quantified spectrophotometrically at 452 nm in a UV–Visible Agilent 8453 spectrophotometer (Agilent Technologies, Waldbronn, Germany), using a standard calibration curve of the drug prepared in DMF. Finally, EE% and DL% were calculated according to the following equations:EE (%) = (W_loaded drug_/W_total drug_) × 100 (1)DL (%) = (W_loaded drug_/W_lipid_) × 100 (2)
where W_total drug_ is the amount of drug included during the formulation process, W_loaded drug_ is the incorporated drug calculated based on the CFZ concentration in the supernatant after NLCs breaking down, and W_lipid_ is the weight of the lipid matrix.

### 2.4. Database Modeling by Artificial Intelligence Tools

The database obtained after CFZ-loaded NLCs formulation and characterization was modeled with the Artificial Intelligence software FormRules^®^ v4.03 (Intelligensys Ltd., London, UK). FormRules^®^ v4.03 is a Neurofuzzy Logic (NFL) software combining Fuzzy Logic and ANNs, that allows us to answer “WHAT-IF” questions by generating sets of “IF-THEN” rules that explain the effect of composition and formulation conditions (inputs) on the properties (outputs) of the obtained NLCs (Appendix A). These “IF-THEN” rules are composed by an antecedent part, relative to inputs, and a consequent part relative to outputs [22]. In order to generate these rules, the software categorizes the different inputs and outputs as high, mid, or low, assigning them a degree of membership [22]. An example of categorization performed by the software for the liquid lipid input is included in Appendix A. The inputs analysed in this study included LL, Tween^®^ 80, lecithin, and CFZ proportions, as well as the homogenization speed during the formulation process. The characterized outputs were particle size, polydispersity index (PdI), zeta potential (ZP), encapsulation efficiency (EE%), and drug loading (DL%).

The training parameters employed to obtain the NFL models were the following: ASMOD (Adaptive Spline Modeling of Data) algorithm, two set densities, structural risk minimization as model selection criteria (C1 = 0.68–0.8 and C2 = 4.8), together with a maximum of 15 nodes per input and a maximum of 2 inputs per submodel. Furthermore, the quality of the predictive models obtained for each NLCs parameter was evaluated through an analysis of variance (ANOVA) and using the determination coefficient of the Training or Test sets R^2^, expressed as a percentage. The ANOVA reflects the accuracy of the models, while R^2^ values are representative of their predictive capacity and were calculated as follows [23,24]:(3)$$R^{2}=[1-\sum_{i=1}^{n}\left(yi-yi\prime \right)2/\;\sum_{i=1}^{n}{\left(yi-yi\prime \prime \right)}^{2}]\times 100$$
where *y* is the experimental value of the output, *y*′ is the output value predicted by the model, and *y*″ is the mean output value. In this way, R^2^ values above 70% are indicative of a good model predictive capacity [25].

Finally, to evaluate the model’s predictive capability, the characteristics of a formulation developed following the NFL guidelines, intended to produce NLCs with optimal properties in terms of small particle size, narrow polydispersity index, negative zeta potential, and high drug payload, were compared to those of a formulation not recommended by the NFL models.

### 2.5. In Silico Docking Analysis

In silico docking analysis was conducted to further investigate the interaction between CFZ and the lipid matrix components. For this purpose, the two-dimensional (2D) molecular structures of CFZ, Precirol^®^ ATO 5, and oleic acid were designed using ChemDraw Professional version 17 (PerkinElmer, Waltham, MA, USA). These structures were then transformed into three-dimensional (3D) conformations using Chem3D version 17, followed by minimization using the Merck Molecular Force Field 94 (MMFF94, Merck, Rahway, NJ, USA) to find the most stable energetic conformation. The optimized 3D structures were saved in .mol file format and subsequently converted to .pdb format using PyMOL version 2.4.0 (Schrödinger, Palo Alto, CA, USA).

Docking simulations were conducted with AutoDock Tools version 4.2.6 (Molecular Graphics Laboratory, La Jolla, CA, USA). A grid box of 120 × 120 × 120 points with a default spacing of 0.375 Å was set around the “receptor” and “ligand”. The docking parameters were set as default using a GA runs number of 50 and Lamarckian genetic algorithm for docking calculation. The binding free energy (ΔG) and inhibition constant (Ki) were calculated to evaluate interaction strength, following previously reported methods [26,27]. Thus, for each docked complex, the conformation exhibiting the lowest free energy of binding (ΔG binding) and affinity constant (Ki) values were selected for further analysis. Final docking poses and the established molecular interactions were visualized and analysed using BIOVIA Discovery Studio v20.1.0.19295 (Dassault Systèmes, Paris, France).

### 2.6. Cell Viability Studies

The cytotoxicity of CFZ-loaded NLC formulations in murine fibroblasts (BALB/3T3) was evaluated though the Alamar Blue assay. This assay is based on the reduction of resazurin (a non-fluorescent dye) to resorufin (a fluorescent compound) by viable cells [28]. Cells were cultured in DMEM supplemented with 10% FBS and 1% antibiotic/antimycotic in standard cell culture conditions (37 °C, 5% CO_2_). The day before the experiment, cells were seeded at 10^5^ cells/mL in 96-well plates. Afterwards, the different samples (blank NLCs and CFZ-loaded NLCs) were added to the monolayers to achieve a final concentration of 60, 40, and 30 µg/mL of solid mass per volume. Furthermore, free CFZ at an equivalent concentration was also evaluated as drug control. To this end, the drug was initially dissolved in DMSO and, afterwards, diluted in complete cell culture media to obtain the desired concentrations. Formulation 10, depicting the best physicochemical properties, was selected to perform the experiments. After 24 h of treatment, the media was removed, and the Alamar Blue working reagent was prepared according to the manufacturer’s instructions and added to each well. After 2 h of incubation, fluorescence (ex. 540–10 nm, em. 580–10 nm) was determined in a microplate reader (Model 680, Bio-Rad, Hercules, CA, USA). Finally, cell viability was calculated according to the following equation where control absorbance corresponds to untreated cells:Cell viability (%) = (Sample Absorbance/Control Absorbance) × 100 (4)

### 2.7. Microbiology Studies

The antimicrobial activity of CFZ-loaded NLCs was assessed by a disk diffusion test, a well-established procedure to determine the resistance of bacteria to antimicrobial compounds [29]. For this purpose, three different *Staphylococcus aureus* strains were used: methicillin-susceptible *S. aureus* (ATCC 25923), methicillin-resistant *S. aureus* (ATCC 1556, USA300 clone), and multidrug-resistant *S. aureus* (BAA-43, Brazilian clone). Bacteria were inoculated in Mueller-Hinton agar plates after adjusting the turbidity to 0.5 on the McFarland scale (1.5 × 10^8^ CFU/mL, approximately). Afterwards, sterile filter disks were impregnated with 20 µL of blank or CFZ-loaded NLCs and placed on the agar plates. Finally, the plates were incubated for 24 h at 35 ± 2 °C. After the incubation period, the generated bacterial inhibition halos were measured to assess the sensitivity of the different microorganisms to the NLC formulations. All the samples were evaluated in triplicate. The experiments were performed using blank and CFZ-loaded Formulation 10.

## 3. Results and Discussion

### 3.1. Clofazimine-Loaded NLC Characterization

In this work, CFZ-loaded NLCs were prepared, using composition and operation conditions previously established, using AI, as a design space for RFB-loaded NLCs [11]. As derived from Table 2, the drug-loaded nanoparticle formulations displayed a wide variability of sizes and PdI values, ranging from 127 ± 11 to 1503 ± 267 nm and from 0.16 ± 0.01 to 0.65 ± 0.40, respectively. On the other hand, zeta potential values were negative in all cases, in the range of −28 ± 0 and −40 ± 2 mV. Furthermore, all formulations showed a suitable capacity to encapsulate CFZ with EE% and DL% values ranging from 80.2 ± 15.4 to 105.3 ± 16.8% and from 2.2 ± 0.1 to 5.0 ± 0.0%, correspondingly.

The overall remarkable drug-loading ability of NLC formulations might be related to the high lipophilicity of CFZ (log P = 7.13) [30], which promotes drug solubility in the lipid phase. This high drug-loading capacity also enables the nanocarriers to act as solubilizing reservoirs, maintaining the drug dispersed and bioavailable in aqueous environments at concentrations up to 1.5 mg/mL. Considering the low aqueous solubility of clofazimine [31], drug solubilization by NLC formulations is expected to have a considerable impact over clofazimine’s bioavailability.

### 3.2. Database Modeling by Artificial Intelligence Tools

The database obtained after nanoparticle formulation and characterization was modeled by FormRules^®^ v4.03, an NFL software. This approach allows us to evaluate the possible interactions between the formulation parameters impacting NLC characteristics, enabling a better understanding of the drug-loaded systems. Figure 1 shows the influence of the different inputs (LL, Tween^®^ 80, lecithin, speed, and CFZ) on the analyzed outputs (size, PdI, ZP, EE%, and DL%), while Table 3 includes the quality parameters of the FormRules^®^ models. FormRules^®^ was able to successfully model the analyzed inputs, with R^2^ values above 70% in all cases, indicative of a suitable prediction capacity [23]. Furthermore, except for PdI, calculated f values were found to be higher than the critical ones for the models’ degrees of freedom, suggesting a suitable accuracy and, in general, a good model performance [23].

Particle size is a relevant characteristic as it determines factors such as interaction with biological systems and drug release [32,33]. The obtained models indicate the variability of this parameter is conditioned by two different inputs: the percentage of lecithin and the LL proportion (Appendix A). The use of a mid LL proportion, in the range of 48–67% approximately, promotes the obtention of nanoparticles displaying a larger size. However, NLC formulations including LL percentages either below 48% or above 67% showed a reduced particle size. The decrease in particle size associated with a high LL proportion is a common phenomenon related either with an inhibition of SL crystallization, or with a decrease in the lipid matrix viscosity, due to the presence of a high amount of oil [34,35]. Conversely, the reduced particle size observed with the use of a high SL amount is not as common and could be associated with the existence of interactions between the lipid matrix components in the presence of the drug. As an example, the use of a 57.5% LL proportion leads to nanoparticles of 594 ± 189 nm (Formulation 6). On the other hand, NLCs prepared with 75% LL and the same amount of lecithin displayed a reduced particle size of 127 ± 11 nm (Formulation 10). Similarly, the use of a 40% LL (60% SL) enables the obtention of NLCs showing a small particle size of 211 ± 7 nm (Formulation 15).

Interestingly, the same dual effect over particle size was observed for lecithin. The use of a mid lecithin percentage, in the range of 0.25–0.75%, promotes NLC formulations having a high particle size. Nonetheless, lecithin incorporation in either lower or higher proportions, below 0.25% or above 0.75%, respectively, enables the development of drug-loaded particles having a reduced particle size. The higher particle size observed when mid proportions of lecithin are employed might be associated with an increase in the total lipid content after the incorporation of a moderate amount of phospholipid. This phenomenon is in line with previous works in the field reporting NLCs’ particle size tends to increase with the use of higher lipid concentrations [36]. The abrupt decrease in particle size observed after further addition of lecithin might be due to a surface tension reduction triggered by the surfactant. Indeed, when a sufficient amount of surfactant is located at the nanoparticle interface, it facilitates droplet division during homogenization [37], counteracting the effects derived from the higher lipid content. As an example, the preparation of drug-loaded NLCs using 0.5% lecithin leads to formulations having a high particle size, 1503 ± 267 nm (Formulation 1), whereas nanoparticle size decreases until 395 ± 83 nm when no lecithin is added, while keeping the remaining formulation parameters constant (Formulation 7). Furthermore, Figure 2A illustrates the individual effects of both LL and lecithin proportions on CFZ-loaded NLCs’ size. Accordingly, the use of a 75% LL together with 1% of lecithin would allow us to develop drug-loaded particles displaying a small particle size of around 89 nm.

Furthermore, particle size distribution and PdI can play a pivotal role in the development of pharmaceutical-grade formulations [38]. These parameters can affect stability, processability, appearance, and product performances [38]. According to the obtained NFL models, PdI values of CFZ-loaded NLCs are conditioned by the combined effects of (1) Tween^®^ 80 and lecithin concentrations; (2) CFZ and lecithin percentage; or (3) LL percentage and stirring speed (Appendix A). Regarding the effect of emulsifier proportions on PdI values, the use of a low amount of Tween^®^ 80 or lecithin, below 2% and 0.5%, respectively, leads to nanoparticles exhibiting a high PdI. This might indicate the emulsifiers’ proportions are not enough to efficiently stabilize the particles, leading to NLCs having a non-uniform size. Conversely, the combination of both emulsifiers including either high Tween^®^ 80 and low lecithin percentages (above 2% and below 0.5%, respectively), or low Tween^®^ 80 and high lecithin proportions (below 2% and above 0.5%, correspondingly), enables the obtention of monodisperse nanoparticles suggesting optimal surface coverage and nanoparticle stabilization. In particular, the use of 1% of Tween^®^ 80 without lecithin (Formulation 7) leads to particles having PdI values of 0.52 ± 0.09. Alternatively, NLC formulations prepared with 3% Tween^®^ 80 showed a decreased PdI of 0.35 ± 0.12 (Formulation 14), when the same amount of lecithin, LL, CFZ, and stirring speed were employed. Interestingly, the use of both high Tween^®^ 80 and high lecithin proportions leads to CFZ-loaded NLCs showing high PdI values. This effect might be associated with a large excess of emulsifier relative to the lipid content. This emulsifier excess is likely to incorporate into the particles, leading to bridging and coalescence [39], and, therefore, increasing formulation polydispersity.

Concurrent effects of CFZ and lecithin percentage were also found to condition CFZ-loaded NLCs polydispersity. If a reduced amount of CFZ (below 3.75%) is added, the use of low lecithin proportions, below 0.5%, would favour the obtention of particles having a low PdI. Conversely, when a high proportion of drug is included in the formulations, a lecithin percentage above 0.5% is required to obtain monodisperse nanosystems. These outcomes might be associated with the formation of large-sized particles led by increases in drug content [40]. This particle size increase enhances the interfacial area that should be stabilized by an increased amount of emulsifier. On the other hand, if a CFZ below 3.75% is used, the incorporation of a lecithin percentage above 0.5% triggers a clear increase in particle size. This finding is in line with previous studies reporting that the addition of an excess of emulsifier increases PdI values considerably, due to an irregular NLCs size reduction [41]. As an example, in Formulation 15, a 5% CFZ together with 1% of lecithin was employed during the formulation process, leading to a narrow PdI of 0.24 ± 0.01.

In addition, PdI values are also influenced by combined effects of LL proportion and homogenization speed. According to the NFL models, if a high LL proportion (above 58%) is used, a mild homogenization speed (below 17,000 rpm) should be employed to obtain NLCs having a reduced PdI. Conversely, NLC matrices incorporating a reduced amount of LL required a vigorous stirring speed to achieve monodisperse formulations. This effect might be attributed to an improved emulsification of the lipid matrix when an increased amount of oil is introduced [35], reducing the required speed to efficiently homogenize the nanoparticle system. Noteworthy, the increase in PdI observed for oil-rich NLCs with the use of stirring speeds above 17,000 rpm could be associated with the presence of an excess of kinetic energy that might promote aggregation [42]. Experimental data showed the use of 75% LL, together with a gentle homogenization speed of 13,400 rpm, leads to NLCs with a reduced PdI of 0.17 ± 0.01 (Formulation 10). However, if the LL proportion is decreased to 40%, while maintaining drug and emulsifier proportions constant, PdI increases until 0.24 ± 0.01, although the higher stirring speed employed (16,800 rpm) contributes to maintain this parameter within suitable ranges (Formulation 15).

Surface charge is another parameter that should be carefully controlled, as it highly affects the nanoparticles’ stability during storage, and also their biological interaction with the human body [33,43]. ZP variability is explained by two different submodels, including a single effect of LL proportion and the interaction between Tween^®^ 80 and lecithin. According to the “IF-THEN” rules gathered from NFL models (Appendix A), the incorporation of a high amount of LL within the NLCs matrix, above 58%, leads to nanoparticles having a highly negative surface charge. As an example, the use of 40% LL leads to nanoparticles having a surface charge of −35 mV (Formulation 15), while formulations prepared maintaining the same emulsifiers and drug content, but increasing the LL proportion to 75%, displayed −39 mV (Formulation 10). This reduction in ZP when a high amount of LL is included in the formulations can be easily associated with the negative charge of oleic acid [44].

On the other hand, the influence of the emulsifiers over ZP is probably related to their ability to interact at the nanoparticle interface [45]. Accordingly, the addition of the non-ionic emulsifier Tween^®^ 80 above ~2% triggers an increase in ZP, regardless of the amount of lecithin used. This phenomenon has been related with the ability of non-ionic emulsifiers, such as Tween^®^ 80, to cover nanoparticles’ diffuse layer, thus altering the measured surface charge and masking the negative charge of the lipid matrix [46]. Conversely, the other emulsifier employed, Epikuron^®^ 145V, is a de-oiled phosphatidylcholine-enriched soybean lecithin, that displays negative charge at neutral pH [47]. However, the amount included in the formulations is not sufficient to counteract the effect produced by the non-ionic emulsifier. In agreement with this, Formulation 2, incorporating a 3% of Tween^®^ 80, shows a ZP of −28 ± 0 mV. Contrarily, if the non-ionic emulsifier proportion is reduced to 1%, while maintaining the same lipid matrix composition and lecithin proportion, a highly negative surface charge, −38 ± 0 mV, is obtained (Formulation 8).

Concerning drug payload in NLC formulations, the “IF-THEN” rules showed EE is conditioned by combined effects of homogenization speed and lecithin%, and also by single effects of LL and Tween^®^ 80 proportions (Appendix A). The use of a high lecithin proportion, above 0.75%, leads to high drug EE values under any homogenization speed. However, when a mid lecithin proportion (0.25–0.75%) is employed, a speed above ~17,000 rpm is required to obtain suitable EE. Furthermore, if a lecithin proportion below 0.25% is used, EE values were reduced, regardless of the homogenization speed employed. The lecithin ability to boost drug encapsulation is in line with previous studies, which proved that the presence of a high amount of lecithin enhances loading capacity of the lipid matrix [48]. Regarding homogenization speed, a suitable balance between emulsifier concentration and stirring speed is essential to promote particle formation and efficient drug incorporation within the lipid matrix. Lecithin is a widely used internal phase emulsifier [49]; the use of low proportions of this compound during the formulation process would increase the mixing speed requirements to obtain particles with suitable characteristics. As an example of lecithin concentration and stirring speed influence over EE, the use of a mid lecithin concentration (0.5%) together with a slow stirring speed (13,400 rpm), leads to a CFZ EE of 89.7 ± 3.2% (Formulation 1). On the other hand, the use of a lecithin concentration above 1% and a moderate stirring speed of 16,800 rpm enables a complete drug encapsulation (EE of 100.0 ± 0.5%), when using the same amounts of Tween^®^ 80, CFZ, and LL (Formulation 15).

Moreover, the LL concentration in the nanoparticle matrix also affected CFZ EE. NLCs formulations including LL percentages either below 48% or above 67% showed a high drug EE, whereas mid LL values, falling between the previous ones, tended to reduce CFZ encapsulation. Furthermore, LL concentrations suitable to promote drug encapsulation are the same as the ones required to obtain small nanoparticles. The increase in EE observed after the introduction of a high LL proportion is also a common phenomenon, related with the increased solubility of most hydrophobic drugs in oils [50]. On the contrary, the high EE observed in SL-enriched formulations is less frequent and might be related with specific interactions between the lipid matrix and the drug. Finally, regarding Tween^®^ 80 proportion influence over drug EE, the NFL models suggested all the evaluated emulsifier proportions (1.5–3%) enabled the formation of particles exhibiting a high CFZ EE.

Alternatively, DL values are affected by single effects of both CFZ concentration and stirring speed during the formulation process (Appendix A). According to the obtained NFL models, the use of a high drug amount, above 4.4%, leads to high DL. This effect can be easily associated with the high hydrophobicity of CFZ [30], that makes it unlikely to remain in the aqueous phase, being therefore incorporated almost exclusively within the lipid matrix upon addition. Moreover, the use of a mid-stirring speed, in the range of 15,200–18,800 rpm, tends to increase DL values. Interestingly, higher stirring speeds tend to reduce the CFZ-loading capacity of NLC formulations. This might be associated with the high kinetic energy generated during the process, which could promote lipid matrix rupture and drug leaching into the external aqueous phase [42], reducing DL values. The obtained experimental data showed the use of a slow homogenization (13,400 rpm) to prepare NLC formulations containing a 3.75% CFZ leads to DL values of 3.0 ± 0.7% (Formulation 13). Notably, if a mid-stirring speed of 16,800 rpm is used instead, while maintaining all the remaining formulation parameters constant, DL values of 3.5 ± 0.2 are achieved (Formulation 12). Additionally, Figure 2B was included to illustrate the effect of both CFZ proportion and homogenization speed on DL. The use of 5% of CFZ, together with a homogenization speed of 16,200 rpm, would produce NLCs displaying DL values around 4.7%, approximately.

### 3.3. Analysis of NFL Model Performance

To analyze NFL models’ performance, one formulation following the model’s guidelines, Formulation 16, was prepared and characterized. Additionally, a formulation prepared using non-recommended parameters, while keeping the amount of CFZ constant, was also developed (Formulation 17). In summary, the IF-THEN rules derived from the NFL models suggest that CFZ-loaded NLC formulations with optimal properties (small particle size, low polydispersity index, negative zeta potential, and high drug payload) can be achieved through the combination of (1) high LL proportion (above 67%); (2) high amount of lecithin (above 0.75%); (3) low Tween^®^ 80 proportion (below 2%); (4) mid-homogenization speed (up to 17,000 rpm); and (5) high CFZ amount (above 4.4%). Hence, the composition and operation conditions used to elaborate Formulations 16 and 17 can be found in Table 4.

The formulation parameters suggested by the NFL models closely align with those employed in Formulation 10, leading to NLCs with a size of 127 ± 11 nm, a PdI of 0.17 ± 0.01, a ZP of −39 ± 1 mV, together with EE and DL values around 100% and 5%, respectively. Similarly, Formulation 16, prepared according to NFL guidelines, showed a small particle size of 132 ± 4 nm and a reduced PdI of 0.17 ± 0.01 (Table 5). Conversely, Formulation 17, developed using parameters not recommended, displayed a high particle size of 570 ± 101 nm, together with a broad PdI of 0.62 ± 0.05. Furthermore, both formulations showed a negative zeta potential and a suitable capacity to encapsulate CFZ, although EE values were found to be slightly higher in Formulation 16. In this way, the optimal characteristics exhibited by both Formulation 10 and 16, prepared according to the IF-THEN rules, enabled validation of the obtained NFL models.

The results strongly suggest that the previously established design for rifabutin (RFB)-loaded nanoparticles can be successfully applied to the development of CFZ-loaded NLCs [11]. This is noteworthy, considering the important differences in the physicochemical properties of these two drugs. RFB displays a molecular weight (MW) of 847 g/mol and a partition coefficient (log P of 4.20) [12,32], whereas CFZ is considerably smaller with a MW of 473.4 g/mol and exhibits higher lipophilicity (log P of 7.13) [30]. Therefore, it can be hypothesized that this validated design space serves as a robust platform for the rapid development of NLC formulations incorporating a wide range of poorly soluble compounds.

### 3.4. In Silico Docking Analysis

To further investigate the molecular interaction between CFZ and lipid excipients, and how these can influence nanoparticle formation and characteristics, an in silico docking study was performed. During a preliminary set of docking studies, the different components present into Precirol^®^ ATO 5 were individually docked with clofazimine to highlight any differences or similarities in the molecular interaction and affinity between the excipient and the drug. As reported by the manufacturer, Precirol^®^ ATO 5 consists of glyceryl esters of palmitic (C16) and stearic (C18) acids, with the diester fraction being predominant. Thus, the formulas reported in Appendix A were taken into consideration. The docking results indicated that the presence of palmitic or stearic acid (or both) in the lipid did not change the affinity of CFZ for the glycerolipid (Figure 3). On the other hand, the presence of the alkyl chain in position 1,3 or 1,2 can change CFZ’s affinity. Indeed, when the substituents are in position 1,2, and consequently, when the chains are closer to each other, CFZ is able to interact with both alkyl chain contemporaneously, enhancing the number of hydrophobic interactions and the affinity (Figure 3 and Appendix A).

Furthermore, docking studies performed with oleic acid evidenced a lower affinity of CFZ for it (1.8 mM) when compared to Precirol^®^ ATO 5 (~240 µM for 1,3 conformation, and ~110 µM for 1,2 conformation) (Appendix A). Considering the formulation process, the drug may have a better tendency to interact and stay within Precirol^®^ ATO 5 lipids (higher affinity). This finding might explain, the increase in CFZ EE, described by NFL models, in formulations incorporating a high proportion of SL (above 52%). Furthermore, the interaction between CFZ and Precirol^®^ can, in turn, change the interaction between oleic acid and Precirol^®^. Indeed, the affinity of oleic acid to Precirol^®^ increases when CFZ is present, thanks to the establishment of strong molecular interactions (Figure 4). Stronger interaction and affinity can be translatable into a higher compactness of lipid phase of the nanoparticles, thus explaining the reduced particle size of CFZ-loaded particles when high amounts of Precirol^®^ are incorporated within the lipid matrix, as reflected by NFL models.

### 3.5. In Vitro Cell Viability Studies

Cell viability studies were performed in murine fibroblasts, a widely used cell model for the evaluation of skin tolerability of pharmaceutical formulation and medical devices [51]. Considering that *Staphylococcus aureus* is a pathogenic bacterium responsible for a good number of skin and soft-tissue infections [52], the use of a fibroblast cell line is considered to be optimal to evaluate nanocarriers’ safety for this application. Figure 5A shows the cell viability data of fibroblasts after treatment with blank and CFZ-loaded NLCs (Formulation 10) at several concentrations (60, 40, and 30 µg/mL). In addition, cells treated with free CFZ at equivalent drug concentrations (3, 2, and 1.5 µg/mL) were also evaluated (Figure 5B). NLCs displayed good biocompatibility with values above 70% in all cases, the threshold value according to ISO 10993-5 [53]. Therefore, the developed formulations possess a suitable skin tolerability. However, both CFZ-loaded NLCs and the free drug at the highest concentration showed the lowest cell viability.

### 3.6. Microbiology Studies

CFZ-loaded NLCs’ antimicrobial activity was evaluated against three *Staphylococcus aureus* strains: methicillin-susceptible, methicillin-resistant, and multidrug-resistant, using a disk diffusion test. After a 24 h incubation, Formulation 10 consistently produced large inhibition halos. Specifically, halos of 19.67 ± 1.15, 21.33 ± 0.58, and 21.67 ± 0.58 mm were observed for the respective *S. aureus* strains. According to previous works in the field, inhibition halos below 10 mm are representative of bacterial resistance, 10 to 15 mm halos are indicative of moderate sensitivity, while values above 15 mm, such as those obtained for the drug-loaded carriers, often indicate high sensitivity towards antimicrobial compounds [54]. Furthermore, blank NLCs having the same composition as Formulation 10 but without containing the drug were used as a control. The results obtained showed no inhibition halo for any of the *S. aureus* strains under evaluation with the blank nanocarriers, confirming that the CFZ encapsulated within the NLCs matrix is responsible for the formulations’ antimicrobial activity.

## 4. Conclusions

The present work shows the outcomes of the different smart technologies’ application during nanoparticle-based formulation development. First, CFZ-loaded NLCs were successfully developed by applying a design space previously established for RFB-loaded formulations using AI tools, such as NFL. Following formulation characterization, NFL was useful to investigate the existing interactions between NLC components and homogenization conditions, after CFZ introduction. Finally, in silico docking simulations were successfully applied to examine possible interactions and affinity between the drug and the lipid matrix components.

The approach succeeded in developing antibiotic-loaded NLC formulations with optimal properties, including small size, high CFZ payload, suitable biocompatibility, and promising antimicrobial potential. Crucially, this efficacy extends to drug-resistant *Staphylococcus aureus* strains, highlighting the clinical relevance of our findings. This work also significantly enhanced our understanding of the lipid nanocarriers, offering a robust design space for future poorly soluble drugs studies. The implementation of smart strategies during pharmaceutical development of nanoparticle-based therapeutics, such as those described in this manuscript, would enable successfully re-exploring the potential of many established antimicrobial compounds.

## Figures and Tables

**Figure 1 pharmaceutics-17-00873-f001:**
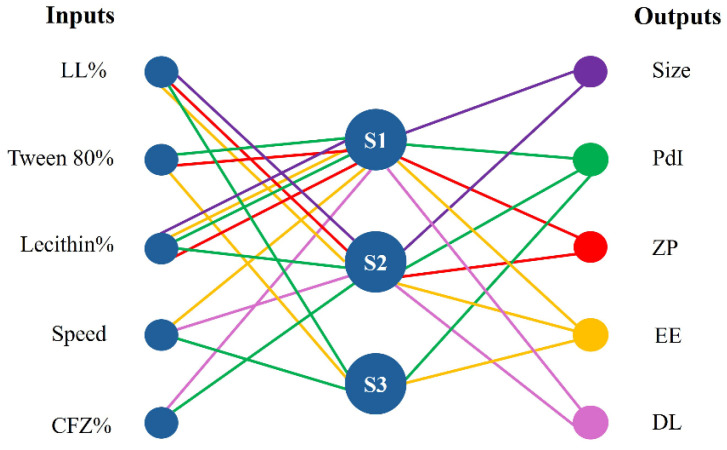
Inputs selected by the NFL models that condition clofazimine-loaded NLC properties (outputs).

**Figure 2 pharmaceutics-17-00873-f002:**
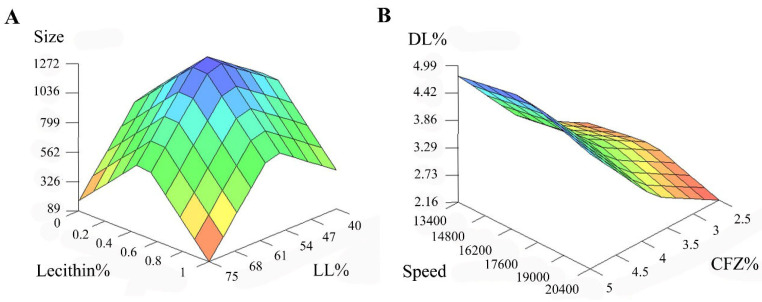
(**A**) 3D plot showing the influence of lecithin and LL over clofazimine-loaded NLCs size. (**B**) 3D plot showing the influence of CFZ and homogenization speed over clofazimine-loaded NLC DL%.

**Figure 3 pharmaceutics-17-00873-f003:**
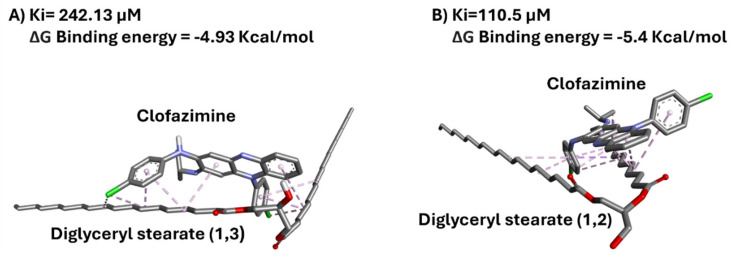
Clofazimine/Diglyceryl stearate binding poses: (**A**) 1,3 and (**B**) 1,2 conformations. In the structures, carbon atoms are shown in grey, oxygen atoms in red, nitrogen atoms in blue, and chlorine atoms in green.

**Figure 4 pharmaceutics-17-00873-f004:**
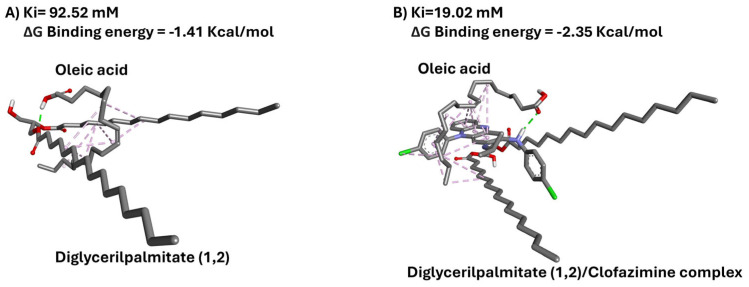
(**A**) Diglyceryl palmitate was set as receptor and oleic acid as ligand. (**B**) Clofazimine/Diglyceryl palmitate complex was set as receptor and oleic acid as ligand. Carbon atoms are displayed in grey, oxygen atoms in red, nitrogen atoms in blue, and chlorine atoms in green.

**Figure 5 pharmaceutics-17-00873-f005:**
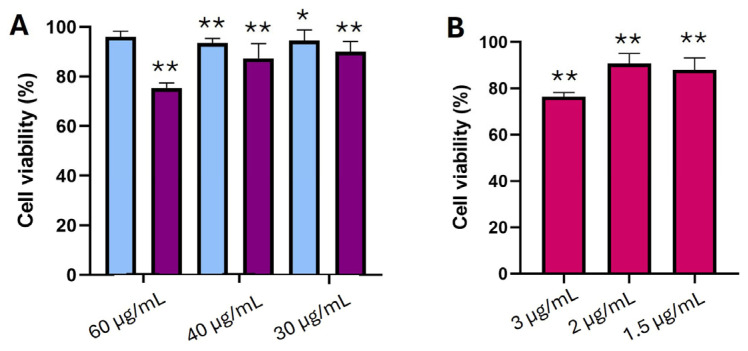
Cell viability (%) relative to positive control (untreated cells) of (**A**) blank (blue color) and CFZ-loaded NLCs (purple color), using several nanoparticle concentrations (60, 40, and 30 µg/mL). (**B**) Free CFZ at the same concentration as in the NLC formulations. Statistically significant differences in cell viability between treated and untreated cells: * *p* < 0.05, ** *p* < 0.01.

**Table 1 pharmaceutics-17-00873-t001:** Clofazimine-loaded NLC manufacturing conditions (design space). * Weight/weight percentages referred to the total weight of the lipid matrix.

Formulation	* LL%	Tween 80% (*w*/*v*)	* Lecithin%	Speed (rpm)	* CFZ %
1	40	1.5	0.5	13,400	5
2	57.5	3	1	16,800	3.75
3	75	1	0	20,400	2.5
4	40	1	0	20,400	5
5	75	3	0.5	13,400	3.75
6	57.5	1.5	1	16,800	2.5
7	40	1.5	0	13,400	5
8	57.5	1	1	16,800	2.5
9	75	3	0.5	20,400	3.75
10	75	1.5	1	13,400	5
11	40	3	0	20,400	2.5
12	57.5	1	0.5	16,800	3.75
13	57.5	1	0.5	13,400	3.75
14	75	3	0	20,400	2.5
15	40	1.5	1	16,800	5

**Table 2 pharmaceutics-17-00873-t002:** Characteristics of the different CFZ-loaded NLC formulations obtained following the composition and manufacturing conditions specified in Table 1.

Formulation	Size (nm)	PdI	ZP (mV)	EE%	DL%
1	1503 ± 267	0.54 ± 0.22	−30 ± 0	89.7 ± 3.2	4.6 ± 0.3
2	630 ± 31	0.58 ± 0	−28 ± 0	99.6 ± 36.4	3.7 ± 1.3
3	353 ± 44	0.52 ± 0.09	−39 ± 1	93.9 ± 3.3	2.4 ± 0.1
4	621 ± 86	0.64 ± 0.02	−31 ± 2	89.3 ± 24.7	4.5 ± 1.2
5	321 ± 134	0.31 ± 0.10	−34 ± 1	84.5 ± 0.9	3.2 ± 0.1
6	594 ± 189	0.54 ± 0.01	−36 ± 1	93.4 ± 1.8	2.6 ± 0.4
7	395 ± 83	0.52 ± 0.14	−31 ± 0	98.2 ± 3.1	5.0 ± 0.0
8	1230 ± 28	0.58 ± 0.05	−38 ± 0	96.4 ± 6.1	2.3 ± 0.2
9	724 ± 619	0.65 ± 0.40	−34 ± 3	105.3 ± 16.8	2.7 ± 0.2
10	127 ± 11	0.17 ± 0.01	−39 ± 1	101.4 ± 10.2	4.9 ± 0.6
11	145 ± 2	0.16 ± 0.01	−31 ± 0	91.3 ± 12.3	2.3 ± 0.3
12	1254 ± 231	0.49 ± 0.44	−40 ± 2	92.1 ± 4.2	3.5 ± 0.2
13	1333 ± 279	0.46 ± 0.51	−36 ± 2	80.2 ± 15.4	3.0 ± 0.7
14	297 ± 102	0.35 ± 0.12	−32 ± 1	86.7 ± 3.6	2.2. ± 0.1
15	211 ± 7	0.24 ± 0.01	−35 ± 0	100.0 ± 0.5	5.0 ± 0.0

**Table 3 pharmaceutics-17-00873-t003:** NFL submodels including parameters modulating CFZ-loaded NLC properties, together with their quality parameters. The most important submodel is bolded.

Output	Submodels	Inputs	R^2^	Calculated f Value	Degrees of Freedom	f Critical (*p* < 0.05)
Size	Submodel 1	Lecithin%	70.34	4.27	5 and 9	3.48
	**Submodel 2**	**LL%**				
PdI	Submodel 1	Tween% × Lecithin%	92.67	5.06	10 and 4	5.96
	**Submodel 2**	**CFZ% × Lecithin%**				
	Submodel 3	LL% × speed				
ZP	**Submodel 1**	**Tween% × Lecithin%**	90.45	7.26	5 and 9	3.48
	Submodel 2	LL%				
EE%	**Submodel 1**	**Speed × Lecithin%**	90.45	5.26	9 and 5	4.77
	Submodel 2	LL%				
	Submodel 3	Tween%				
DL%	**Submodel 1**	**CFZ%**	97.06	59.42	5 and 9	3.48
	Submodel 2	Speed				

**Table 4 pharmaceutics-17-00873-t004:** Manufacturing conditions employed to prepare Formulations 16 and 17. * Weight/weight percentages refer to the total weight of the lipid matrix.

Formulation	* LL%	Tween 80% (*w*/*v*)	* Lecithin%	Speed (rpm)	* CFZ %
16	75	1.5	1	15,200	5
17	55	1	0.4	13,400	5

**Table 5 pharmaceutics-17-00873-t005:** Experimental properties of CFZ-loaded NLC Formulations 16 and 17.

Formulation	Size (nm)	PdI	ZP (mV)	EE%	DL%
16	132 ± 4	0.17 ± 0.01	−22 ± 1	92.6 ± 0.2	4.6 ± 0.2
17	570 ± 1	0.62 ± 0.05	−24 ± 0	84.1 ± 13.1	4.3 ± 0.8

## Data Availability

Data is contained within the article or Appendix A.

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
