# Peer review of "Design of Clofazimine-Loaded Lipid Nanoparticles Using Smart Pharmaceutical Technology Approaches"

_pharmaceutics, 2025, doi:10.3390/pharmaceutics17070873_

Round 1
Reviewer 1 Report
Comments and Suggestions for Authors
The manuscript describes the formulation of clofazimine-loaded nanostructured lipid carriers using a previously defined design space and analyzes parameter impact through Neurofuzzy Logic modeling. While the formulation and characterization are well-conducted, the absence of any biological validation or new hypothesis testing limits its translational value. It lacks critical experimental validation (especially biological activity and model verification). Only after adding in vitro efficacy data and addressing key analytical and modeling issues should it be considered for publication.
- The design space is adapted from a previous study using rifabutin without proper justification for its direct application to clofazimine, which has different physicochemical properties. The authors need to provide more rigorous justification or include a Design of Experiment (DoE) tailored for CFZ to validate that the transferred design space is applicable.
- While the NFL approach gives insight into parameter influence, no experimental validation of NFL-predicted optimal formulations is shown. The authors are advised to formulate at least one or two new formulations based on NFL predictions and compare their characteristics experimentally to validate the model.
- In Table 2, some formulations (specifically Formulations 9 and 10), are shown to have % EE as 105.3 and 101.4, which are practically not possible and not physically meaningful. The authors have to re-examine and recalibrate quantification methods or clearly explain if this is due to standard curve error, matrix interference, or incomplete dialysis.
- The manuscript presents formulation and in vitro characterization only. There is no in vitro antimicrobial efficacy or in vivo pharmacokinetic/therapeutic study, making the therapeutic advantage of the NLCs purely speculative. The authors can include biological testing, such as MIC (minimum inhibitory concentration) against relevant microbial strains, or basic in vivo pharmacokinetic profiling, to demonstrate improved performance over free CFZ.
Reviewer 2 Report
Comments and Suggestions for Authors
In this study, a combination of smart pharmaceutical technology approaches was used to fabricate nanostructured lipid carriers to improve the water solubility and bioavailability and safety of clofazimine (CFZ). Before being accepted, many issues need to be explained in detail.
- To what extent can the liposomes prepared in this paper improve the water solubility of CFZ?
- Characterizations (such as, size distribution, morphology, etc) on the fabricated CFZ-NLCs should be provided;
- Data presented in Table 1 (CFZ%) should be provided as the format of mean ± SD;
- The release profile of CFZ from the CFZ-NLCs should be studied;
Round 2
Reviewer 1 Report
Comments and Suggestions for Authors
The authors have adressed and resolved all the concerns raised in the first round of revision. Hence, the present form of manuscript seems appropriate for acceptance.
Reviewer 2 Report
Comments and Suggestions for Authors
Accept in present form